# Appendectomy and Non-Typhoidal *Salmonella* Infection: A Population-Based Matched Cohort Study

**DOI:** 10.3390/jcm10071466

**Published:** 2021-04-02

**Authors:** Den-Ko Wu, Kai-Shan Yang, James Cheng-Chung Wei, Hei-Tung Yip, Renin Chang, Yao-Min Hung, Chih-Hsin Hung

**Affiliations:** 1Institute of Biotechnology and Chemical Engineering, I-Shou University, Kaohsiung 840, Taiwan; dkw3275@gmail.com; 2Department of Emergency Medicine, Kaohsiung Veterans General Hospital, Kaohsiung 813, Taiwan; 3School of Post-Baccalaureate Medicine, Kaohsiung Medical University, Kaohsiung 807, Taiwan; jenniferyang1993@gmail.com; 4Graduate Institute of Integrated Medicine, China Medical University, Taichung 404, Taiwan; wei3228@gmail.com; 5Division of Allergy, Immunology and Rheumatology, Institute of Medicine, Chung Shan Medical University, Taichung 402, Taiwan; 6Management Office for Health Data, China Medical University Hospital, Taichung 404, Taiwan; fionyip0i0@gmail.com; 7Department of Internal Medicine, Kaohsiung Municipal United Hospital, Kaohsiung 804, Taiwan; 8College of Health and Nursing, Meiho University, Pingtung 912, Taiwan; 9Department of Internal Medicine, Institute of Medicine, Chung Shan Medical University, Taichung 402, Taiwan

**Keywords:** appendectomy, non-typhoidal *Salmonella* infection, NTS, cohort study, National Health Insurance Research Database

## Abstract

The potential association between appendectomy and non-typhoidal *Salmonella* (NTS) infection has not been elucidated. We hypothesized that appendectomy may be associated with gut vulnerability to NTS. The data were retrospectively collected from the Taiwan National Health Insurance Research Database to describe the incidence rates of NTS infection requiring hospital admission among patients with and without an appendectomy. A total of 208,585 individuals aged ≥18 years with an appendectomy were enrolled from January 2000 to December 2012, and compared with a control group of 208,585 individuals who had never received an appendectomy matched by propensity score (1:1) by index year, age, sex, occupation, and comorbidities. An appendectomy was defined by the *International Classification of Diseases, Ninth Revision, Clinical Modification Procedure Codes*. The main outcome was patients who were hospitalized for NTS. Cox proportional hazards models were applied to estimate the hazard ratios (HRs) and 95% confidence intervals (CIs). Two sensitivity analyses were conducted for cross-validation. Of the 417,170 participants (215,221 (51.6%) male), 208,585 individuals (50.0%) had an appendectomy, and 112 individuals developed NTS infection requiring hospitalization. In the fully adjusted multivariable Cox proportional hazards regression model, the appendectomy group had an increased risk of NTS infection (adjusted HR (aHR), 1.61; 95% CI, 1.20–2.17). Females and individuals aged 18 to 30 years with a history of appendectomy had a statistically higher risk of NTS than the control group (aHR, 1.92; 95% CI, 1.26–2.93 and aHR, 2.67; 95% CI, 1.41–5.07). In this study, appendectomy was positively associated with subsequent hospitalization for NTS. The mechanism behind this association remains uncertain and needs further studies to clarify the interactions between appendectomy and NTS.

## 1. Introduction

Appendectomy is one of the most-commonly performed surgical procedures in the world. A recent meta-analysis of the incidence of appendectomy in Northern America was 100 per 100,000 person years [1], while it was 107.76 in Taiwan [2]. Studies have shown that the appendix may be an important component of human immune function [3,4]. Absence of an appendix has been mentioned in relation to recurrent infection with *Clostridium difficile* [5]. Recently, a study that enrolled patients who underwent incidental prophylactic appendectomy during 2004–2008 showed profound and long-term dysbiosis in these patients, sometimes for years [6]. Reduced microbial diversity may reflect the severity of the disease in critically ill patients and be associated with mortality [7]. Appendectomy might disrupt the immune function and studies have observed the relationship between antecedent appendix removal and the risk of pulmonary tuberculosis and sepsis [8,9]. Global non-typhoidal *Salmonella* (NTS) infection occurs in millions of people annually [10,11,12,13,14]. NTS may cause severe invasive bacteremia or disseminated disease [15,16]. The numbers of host risk factors predispose individuals to NTS [17,18]. These risk factors include the extremes of age [19], diabetes [19], malignancy [20], rheumatologic disease [19,21], use of immunomodulatory drugs [18], transplantation [22], and HIV infection [17,23]. About half a century ago, gastrectomy had been shown to be associated with an increased risk of subsequent NTS infection due to achlorhydria, rapid food emptying and altered intestinal flora [24]. Nowadays, it is widely accepted that when the bacterial population in the gastrointestinal tract is unstable, NTS is more likely to take advantage of the situation and invade the gastrointestinal tract [25]; on the other hand, appendectomy might cause long-term disturbance of the microbiome [6]. We hypothesized that patients who experienced removal of the appendix were susceptible to NTS. This population-based propensity score-matched (PSM) cohort study was conducted to examine the impact of appendectomy on subsequent NTS infections requiring hospital admission.

## 2. Materials and Methods

### 2.1. Data Source

Since 1995, more than 99% of the Taiwan population have been insured through a single-payer National Health Insurance program launched by the government. The medical claims contribute to the National Health Insurance Research Database (NHIRD). Previous studies demonstrated the high validity of data derived from the NHIRD [26]. This study used the hospitalization dataset, which records the disease diagnosis and procedure of therapy received during the admission. The diagnostic codes of the claims are recorded according to the International Classification of Diseases, 9th Revision, Clinical Modification (ICD-9-CM).

### 2.2. Standard Protocol Approvals, Registrations, and Patient Consents

The Research Ethics Committee of China Medical University and Hospital in Taiwan (CMUH104-REC2-115(AR-4)) approved this study. As the data used consisted of the de-identified secondary data set released for research purposes and were analyzed anonymously, the need for informed consent was waived.

### 2.3. Study Subjects

An appendectomy was defined according to the ICD-9-CM procedure code 47. The appendectomy group consisted of 208,585 individuals ages 18 and over with a newly received appendectomy from 1 January 2000, through 31 December 2012; individuals who received an appendectomy from 1997 to 1999 were excluded. To minimize confounding from other alimentary surgical procedures, individuals who received a gastrectomy (ICD-9-CM procedure code 43.5-43.9), cholecystectomy (ICD-9-CM procedure code 51.2), or intestinal or large bowel resection (ICD-9-CM procedure code 45.6-45.9) before the index date or underwent multiple concurrent procedures at the time of appendectomy were excluded. Patients diagnosed with cancer (ICD-9-CM code 140-208) before the index date were also excluded.

As proton pump inhibitors provide a favorable environment for NTS, patients with peptic ulcer disease (a proxy for proton pump inhibitors) before the index date were excluded. Individuals with hospitalized NTS within one month after the index date were also excluded to avoid confounding by the possible effect of perioperative antibiotics. The first date of hospitalization for appendectomy was the index date, and this date was assigned to the accordant matched controls (defined as the first healthcare use occurring in the index year) with the same criteria. Patients having a history of NTS are at risk for recurrent NTS, so those patients were also excluded.

Finally, 208,585 patients with appendectomy without a medical history of NTS before the index date (traced back from 1997 through 1999) were included. To minimize surveillance bias, these exposed participants were compared with the 208,585 sex-, age-, index date-and comorbidity-matched individuals in the non-appendectomy group by propensity score matching (PSM) from the same inpatient dataset. We performed a rematch by greedy algorithm. For each study case with appendectomy, the corresponding comparison case without appendectomy was selected based on the closest propensity score. Propensity scores were calculated using a logistic regression model to calculate the probability of appendectomy assignment and included the following baseline variables: sex, age, occupation, and year of index date. The comorbidities analyzed in the study included hypertension (ICD-9-CM code 401-405), diabetes (ICD-9-CM code 250), hyperlipidemia (ICD-9-CM code 272), coronary artery disease (CAD) (ICD-9-CM code 410-414), cerebrovascular disease (CVD) (ICD-9-CM code 430-438), chronic kidney disease (CKD) (ICD-9-CM code 585), chronic obstructive pulmonary disease (COPD) (ICD-9-CM code 491, 492, 496), human immunodeficiency virus (HIV) (ICD-9-CM code 042), liver cirrhosis (ICD-9-CM code 571.5), and systemic lupus erythematosus (SLE) (ICD-9-CM code 710.0).

Each case in the study and control groups were followed from individual index date until an event (hospitalization for NTS), withdrawal from the NHI program or December 2013. We adopted the PSM method to account for a similar distribution of baseline characteristics between both groups.

### 2.4. Identification of Main Outcome

The outcome was patients with NTS recorded in the hospitalization dataset, a subset of the NHIRD; the incidence of new-onset NTS depends upon the administrative ICD coding of 003.xx [11]. The physician responsible for the patient must make the diagnosis using the appropriate ICD code based on careful evaluation and examination, including analysis of stool and/or blood cultures. The coding system is considered validated as the government periodically audits claims for payment purposes. The fine for fraud is 100 times the amount of the fraudulent claims collected from the NHI Bureau. To control the possible bias due to perioperative antibiotics, individuals experienced NTS within one month of the index date were excluded.

### 2.5. Negative Exposure Control Analysis

Negative control has been used to detect unmeasured confounding. Diverticulitis was selected as an alternative exposure (ICD-9-CM code 562.x), and based on review of current pathophysiological mechanisms, it was not associated with subsequent NTS. Therefore, any association between diverticulitis and subsequent NTS may hint at the presence of unmeasured confounding factors.

### 2.6. Statistical Analysis

The first record of each participant hospitalized for NTS was used to calculate the risk of NTS. The density of NTS events per 10,000 person-years was calculated in both groups. We used PSM to control for sampling bias. The propensity score presented an individual’s probability of developing NTS, and the score was determined by a multivariable logistic regression model. The difference of the baseline characteristics between the study and the comparison group were compared by the standardized mean difference (SMD). A SMD of 0.1 or less indicates a negligible difference between the two groups.

We estimated the crude hazard ratio (HR) and 95% confidence interval (CI) using the univariable Cox proportional hazard model. Variables found to be statistically significant in the univariable model were further examined in the multivariable model. The multivariable Cox proportional hazard model was used to estimate the adjusted HR (aHR), including hypertension, diabetes, CAD, CVD, CKD, COPD, HIV, liver cirrhosis, and SLE. The Kaplan−Meier method was adopted to obtain the cumulative incidence of NTS in the two groups. The log-rank test was utilized to compare the differences between the two groups. All statistical tests were two-sided, and *p* values of 0.05 or less were considered statistically significant.

In the main model, we excluded patients with PUD (a proxy to minimize the effect of proton pump inhibitors utilization) [11,27] before the index date and patients who had NTS within one month of the index date (a proxy to minimize the effect of perioperative antibiotic utilization). To validate the findings in the main model, several post hoc sensitivity analyses by different definitions of enrollment (model 2 to 5) were conducted. Because the effects of antibiotic treatment might be longer lasting than 30 days; in model 2, we excluded patients having NTS infection occurred within 90 days of the index date. In model 3, because antibiotic utilization is likely a significant confounder, and with no prescription information in the hospitalization dataset, we excluded patients having bacterial infection within 6 months before the index date (a proxy to minimize the effect of antibiotic utilization on the participants; ICD-9 codes of bacterial infection are 001-005, 008.1-008.5, 020-027, 030-041, 076, 320, 420.9, 421.0, 422.92, 481-483, 511.1, 522.4-522.7, 523.3-523.5, 527.3, 528.3, 566, 567.0-567.2, 569.5, 572.0, 590, 595.89, 595.9, 597.0, 599.0, 614-616, 680-686, 785.52). In model 4, PUD and other comorbidities related to immunocompromise that conferred increased risk of NTS infection were excluded before the index date. In model 5, PUD and other comorbidities related to immunocompromise that confer increased risk of NTS infection were not excluded and adjusted as covariates in regression analysis.

## 3. Results

### 3.1. Patient Characteristics

Of 417,170 participants (215,221 (51.6%) male) aged 18 years and older, 208,585 individuals (50%) had experienced appendectomy (107,823 male (52%)) and 112 individuals 0.05% developed hospitalized NTS. The 208,585 individuals who did not have appendectomy (107,398 men (51%)) were matched by age, sex, and comorbidities (Table 1). PSM resulted in 208,585 matched individuals in each group. In the study group and comparison group, the baseline characteristics were well balanced. The mean (SD) age was 38.8 (15.2) years in the study group and 40.8 (16.7) years in the control group. The median (SD) follow-up times were 7.29 (3.87) years in the study group and 6.73 (3.54) years in the control group. Individuals in the study group, compared with those in the control group, had similar proportions of occupation and comorbidities but a lower proportion of hypertension (13,982 individuals (7%) vs. 26,344 individuals (13%); SMD, 0.20), diabetes (7646 individuals (3.7%) vs. 12,891 individuals (6.2%); SMD, 0.12), and cerebrovascular disease (3189 individuals (1.5%) vs. 1492 individuals (3.1%); SMD, 0.11). The mean (SD) hospital stay for appendectomy was 5.65 (42.9).

### 3.2. Outcomes

Table 2 shows the results of the univariable and multivariable Cox regression analysis, in which the incidence rate of NTS after appendectomy was 0.74 per 10,000 person-years and that in the comparison group was 0.55 per 10,000 person-years. There were 77 events of hospitalized NTS without appendectomy, and 112 events of hospitalized NTS after undergoing appendectomy. The individuals who had histories of appendectomy were more likely to develop NTS (unadjusted HR, 1.35; 95% CI, 1.01–1.8). The multivariable Cox regression analysis showed a positive association between appendectomy and new-onset hospitalized NTS. After adjusting for demographics, occupation, and comorbidities (except hyperlipidemia) at the baseline, individuals with appendectomy had a 61% increased risk of developing hospitalized NTS than subjects without appendectomy, with an adjusted HR of 1.61 (95% CI, 1.20–2.17). Table 2 also presents the relevant risk factors for NTS infection, and these were age 61–70, 71–80. The 81–100 group with aHR 2.62 (95% CI, 1.57–4.37), 2.51 (95% CI, 1.34–4.60), 4.50 (95% CI, 2.16–9.38), diabetes (aHR, 1.75), COPD (aHR, 2.65), liver cirrhosis (aHR 4.48), HIV (aHR, 22.8), and SLE (aHR, 23.3). Thus, this study proposes appendectomy as a potential novel risk for subsequent NTS infection requiring hospitalization.

### 3.3. Sensitivity Analyses

Table 3 provides five models to examine the stability of HR of hospitalized NTS infection with the different definitions of appendectomy exposure and events of main outcome. In the subset of NHIRD (hospitalization dataset analyzed in this study), there was no prescription information to identify the length of antibiotic treatment. We developed 1, 2, and 3 to minimize the effect of antibiotic utility. The wash-out period in model 2 was up to 90 days. The aHRs were 1.61 (95% CI 1.20 to 2.17), 1.58 (95% CI 1.17 to 2.13) and 1.61 (95% CI 1.20 to 2.16) in models 1, 2 and 3. Furthermore, in model 4, we excluded immunocompromised cases potentially prone to have NTS infection and the aHR was 1.71 (95% CI 1.26 to 2.33) of NTS infection for appendectomy exposure. In model 5, PUD was included into regression analysis and the aHR was 1.24 (95% CI 1.02 to 1.52).

### 3.4. Subgroup Analysis

Table 4 reveals the risk of NTS infection requiring hospitalization in different subgroups. In the age subgroup analysis, in comparison to the age-matched non-appendectomy controls, individuals aged 18 to 30 years with appendectomy had a significantly higher risk of subsequent hospitalized NTS (adjusted HR, 2.67; 95% CI, 1.41–5.07; *p* < 0.01). Prior history of appendectomy in patients age 51–60, 61–70, 71–80, and 81–100 have adjusted HR of 2.68 (95% CI, 0.85–8.47), 1.12 (95% CI, 0.51–2.45), 0.90 (95% CI, 0.34–2.35), and 0.34 (95% CI, 0.07–1.63) respectively. The interaction for the age subgroup was not significant (*p* value for interaction = 0.10). In the sex subgroup analysis, females with a history of appendectomy had an increased risk of NTS infection compared with females without appendectomy (adjusted HR, 1.92; 95% CI, 1.26–2.93; *p* < 0.01), while there was no significant association between appendectomy and risk of NTS infection for male patients. However, the *p* value for interaction was not significant (0.82). In the occupation subgroup analysis, compared to matched participants without appendectomy, individuals with a white-collar occupation had a significantly increased risk of subsequent hospitalized NTS (adjusted HR, 1.76, 95% CI, 1.17–2.65; *p* < 0.01). In the comorbidity-subgroup analysis, in general, compared with matched patients without appendectomy, the risk of hospitalized NTS infection became nonsignificant whenever any one of the comorbidities presented. Appendectomy appeared to have a higher association in the relatively healthy participants in the study (e.g., within all participants without hypertension, subjects in the appendectomy group were at higher risk of NTS infection in comparison to participants without appendectomy; aHR, 1.77; 95% CI, 1.26–2.47; within all participants without diabetes, subjects in the appendectomy group were at higher risk of NTS in comparison to participants without appendectomy; aHR, 1.72; 95% CI, 1.25–2.37; and within all participants without SLE, subjects in the appendectomy group were at higher risk of NTS infection in comparison to participants without appendectomy, aHR, 1.67; 95% CI, 1.24–2.26).

Table 5 displays our analysis stratified by the follow-up years. In the first six months, the relative risk of hospitalized NTS compared with the subjects without appendectomy was 1.83 (95% CI, 0.74–4.53). During the follow up of six months to one year after appendectomy, the relative risk of having hospitalized NTS was 0.68 (95% CI, 0.25–1.89). After >1 year of follow up, the adjusted HR was 1.74 (95% CI, 1.25–2.43).

The alternative exposure (diverticulitis) showed no significant association between diverticulitis and subsequent hospitalizations for NTS (adjusted HR, 0.85; 95% CI, 0.18–3.95) (Table 6). The cumulative incidence curve of NTS in the appendectomy cohort was significantly higher than that in the non-appendectomy group (log-rank test *p*-value = 0.04) (Figure 1).

## 4. Discussion

In this study, a prior appendectomy was associated with a 61% increase of risk of developing hospitalized NTS. This is a novel finding. The link between previous appendectomy and subsequent NTS infection requiring hospitalization has never been discussed or confirmed before. Acute appendicitis occurs predominantly at 20 to 30 years of age with male predominance [28]. Similarly, our study found that about sixty percent of patients receiving an appendectomy were aged <40 years, with a male predominance. Some studies denoted that there might be postoperative change of the microbiome. Change of microbial composition was observed in patients received cholecystectomy [29,30]. Gastrointestinal microbiota showed higher species diversity and richness after gastrectomy in patients with gastric cancer [31]. Some studies have further accessed the interaction between dietary intake, gastric bypass surgery, and the trend of microbial change [32]. The balance of intestinal microbiota is critical to support the resistance against colonization by exogenous microorganisms. NTS was found to be competitive against the microbiome during inflammation in the gut and subsided when the inflammation ceased [33]. Butyrate as a feed additive has been widely used to improve the intestinal health of poultry and reduce the proliferation of *Salmonella* [34]. Appendectomy was reported significantly associated with low levels of butyrate-producing bacteria [35]. Furthermore, in one recent study, authors found that patients who underwent prophylactic appendectomy had lower levels of abundance and diversity of normal gastrointestinal tract species over the long-term [6]. Our findings are in alignment that the relative risk of NTS infections rise to statistically significant after one year.

A previous study demonstrated that risk factors for NTS infection consist of aging and immunocompromise [17], which corresponded with our findings in Table 2 (e.g., diabetes, COPD, liver cirrhosis, SLE, HIV).

The underlying mechanism by which appendectomy is associated with the risk of developing NTS infection remains unclear. First, the appendix contains large amounts of gut-associated lymphoid tissue, which is thought to be involved in immune function. Peyer’s patches, and the appendix, are the sites of antigen sampling and induction in the mucosal immune system [36]. Therefore, an appendectomy might change the immune system. Secondly, the appendix can provide a suitable environment for normal intestinal flora through biofilm formation [37,38]. As a result, an appendectomy may disrupt the gut microbiota configurations subsequently supporting NTS development [39,40].

The post hoc stratified analysis showed that compared with matched non-appendectomy controls, patients who received an appendectomy were associated with an increased risk of NTS especially in the subgroup of females and the subgroup for individuals aged 18 to 30 years. A recent meta-analysis of the global burden of invasive NTS disease did not find a link between sex and the incidence of invasive NTS disease [41]. It is intriguing however that this might not hold true in the context of prior appendectomy. Some animal and human studies have shown that disease patterns and gut microbiota differ by sex [42,43]. We speculate this novel result might be multifactorial, including environmental exposure (females are the main food handlers). However, further studies are needed to examine such discrepancies. Since advanced age is an independent risk factor for NTS infection, we specifically examined the interaction of age between appendectomy and outcome of interest in this study. In the age-subgroup analysis, compared with non-appendectomy controls, the population who received an appendectomy was at risk of new-onset NTS infection at the age of 18–30. The lack of association between appendectomy and NTS infection in the patients >50 years hold true in the elderly patients shown in Table 4. New-onset post-appendectomy-associated NTS infection was higher in patients without underlying diseases. It may be possible to avoid hospitalized NTS in post-appendectomy patients in these subgroups.

Previous literature had described the advantages of using NHIRD in research [44]. These included enormous samples, one single ethnic population, and long-term comprehensive follow-up. We attempted to control the measurable covariates in both groups through PSM. In this study, we have examined and shown that diabetes, COPD, liver cirrhosis, SLE, and HIV infection are highly associated with NTS infection, and this is a kind of positive control analysis indicating the fitness of our models.

Some limitations in this study should be addressed. First, the diagnoses of NTS infection were based on administrative ICD-9-CM codes rather than a bacterial culture. The Bureau of NHI had a regular auditing mechanism. Quarterly expert reviews on random samples of inpatient claims data with a sampling rate of 1 in 10 were performed by the Bureau of NHI to ensure the accuracy. Misclassification bias may have occurred and some of the subgroup analyses where very few events were included may not be relevant. Second, the NHI program began in 1995; medical utilization before 1995 could not be traced. Therefore, the possibility that patients selected in the comparison cohort had undergone surgery before 1995 cannot be completely excluded. However, such a sampling bias would, on the contrary, underestimate the risk of the primary outcome [45]. Third, NHIRD does not provide lifestyle information, such as tobacco use, physical activity, body mass index, diet, and exercise. We have carefully used diabetes, hypertension, and hyperlipidemia as a proxy of metabolic status and COPD for tobacco use. Fourth, there is no detailed information about the route of NTS infection, its specific serotype, and level of disease severity in NHIRD and this is an inherent major limitation. In this study, we recruited patients from the subgroup of hospitalized NHIRD as a proxy for alluding to severe NTS infection.

Despite meticulous statistical analyses for possible confounding factor adjustment, bias may have occurred. We have applied a number of sensitivity analyses to control the measurable confounders and negative exposure controls to examine the unmeasured confounding. These observations suggest that the presence of confounding factors is less likely when assessed from this perspective. Finally, microbial dysbiosis may be a key intermediate between appendectomy and subsequent NTS infection, but that has not been established in the current study for the lack of detailed information regarding the interactions between appendectomy, change of microbiome metabolites (short-chain fatty acids, such as acetate, propionate, and butyrate)**,** and the information of antibiotic use.

## 5. Conclusions

We conclude that Taiwanese residents with a history of appendectomy were associated with a risk of hospitalization for NTS. The risk was significant in women, and individuals aged 18–30 years. A small number of NTS infection diagnoses occurred in the study, thus limiting the conclusions somewhat. Clinicians are advised to implement prudently the post-operation education for patients to get rid of possible NTS contaminated food in the endemic area. It is of note that since this observation study was performed in one, relatively small country, if similar studies were to be done in the future in other countries of non-Asian origin, the results may be exactly the opposite.

## Figures and Tables

**Figure 1 jcm-10-01466-f001:**
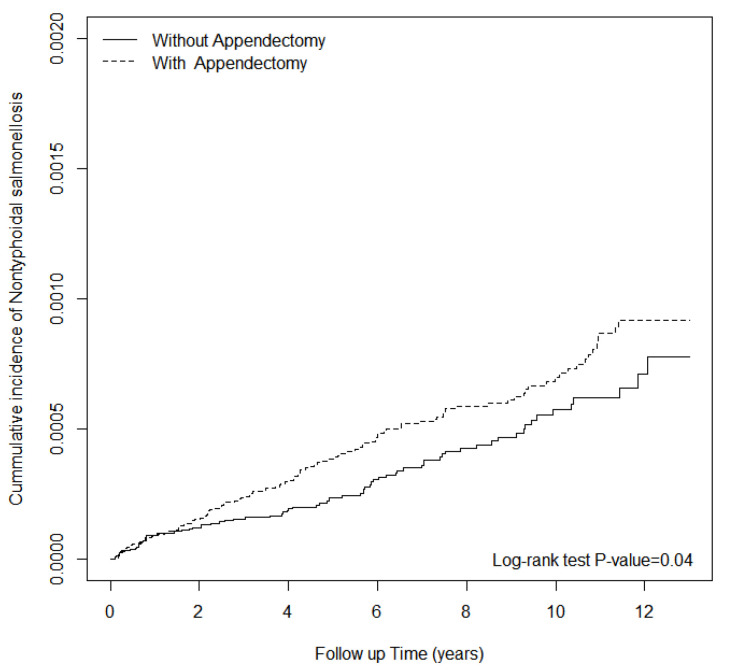
The cumulative incidence of hospitalized NTS infection for patients with and without appendectomy.

**Table 1 jcm-10-01466-t001:** Baseline Patient Characteristics.

	Non-Appendectomy		Appendectomy		
	(N = 208,585)		(N = 208,585)		
Variables	n	%	n	%	SMD
Gender					0.004
Female	101,187	49%	100,762	48%	
Male	107,398	51%	107,823	52%	
Age group					
18–30	67,712	32%	73,158	35%	0.06
31–40	48,032	23%	51,535	25%	0.04
41–50	38,410	18%	38,985	18.5%	0.007
51–60	23,946	12%	22,957	11%	0.02
61–70	14,661	7%	11,934	5.7%	0.05
71–80	11,166	5.4%	7381	3.5%	0.09
81–100	4658	2.2%	2635	1.3%	0.07
mean, (SD)	40.8	(16.7)	38.8	(15.2)	0.13
Occupation					
white-collar worker	109,333	52%	114,108	55%	0.046
blue-collar worker	47,233	23%	48,197	23%	0.011
farmer	4268	2%	4053	2%	0.007
fisher	25,437	12%	22,041	11%	0.05
others	22,314	11%	20,186	10%	0.03
Comorbidities					
hypertension	26,344	13%	13,982	7%	0.20
diabetes	12,891	6.2%	7646	3.7%	0.12
hyperlipidemia	5811	2.8%	3185	1.5%	0.09
CAD	8518	4.1%	3908	1.9%	0.13
CVD	6512	3.1%	3189	1.5%	0.11
CKD	1492	0.7%	762	0.4%	0.05
COPD	3235	1.6%	1532	0.7%	0.08
HIV	105	0.1%	93	0.0%	0.003
Liver cirrhosis	1036	0.5%	551	0.3%	0.04
SLE	506	0.2%	233	0.1%	0.03

Abbreviations: CAD: coronary artery disease; CVD: cerebrovascular disease; CKD: chronic kidney disease; COPD: chronic obstructive pulmonary disease; HIV: human immunodeficiency virus; SLE: systemic lupus erythematosus.

**Table 2 jcm-10-01466-t002:** Hospitalized non-typhoidal *Salmonella* (NTS) infection incidence rate and risk factors.

				Crude Analysis		Adjusted Analysis ^†^	
Variables	Events	PY	IR	HR	95% CI	HR	95% CI
Appendectomy							
No	77	1,402,999	0.55	1.00	(reference)	1.00	(reference)
Yes	112	1,521,600	0.74	1.35	(1.01, 1.8) *	1.61	(1.20, 2.17) **
Gender							
Female	96	1,426,675	0.67	1.00	(reference)		
Male	93	1,497,924	0.62	0.92	(0.69, 1.23)		
Age group							
18–30	49	1,056,867	0.46	1.00	(reference)	1.00	(reference)
31–40	36	706,585	0.51	1.10	(0.72, 1.69)	1.05	(0.68, 1.61)
41–50	31	550,161	0.56	1.22	(0.78, 1.91)	1.15	(0.73, 1.80)
51–60	15	300,584	0.49	1.09	(0.61, 1.95)	0.93	(0.52, 1.68)
61–70	27	168,690	1.60	3.49	(2.18, 5.59) ***	2.62	(1.57, 4.37) ***
71–80	19	109,128	1.74	3.84	(2.25, 6.52) ***	2.51	(1.37, 4.60) ***
81–100	12	32,584	3.68	8.16	(4.32, 15.4) ***	4.50	(2.16, 9.38) ***
Occupation							
white-collar worker	102	1,568,790	0.65	1.00	(reference)		
blue-collar worker	47	668,814	0.70	1.08	(0.77, 1.53)		
farmer	3	59,873	0.50	0.77	(0.24, 2.43)		
fisher	25	331,922	0.75	1.16	(0.75, 1.80)		
others	12	295,200	0.41	0.63	(0.34, 1.14)		
Comorbidities							
hypertension							
No	152	2,714,893	0.56	1.00	(reference)	1.00	(reference)
Yes	37	209,706	1.76	3.21	(2.24, 4.62) ***	1.12	(0.67, 1.85)
diabetes							
No	166	2,813,306	0.59	1.00	(reference)	1.00	(reference)
Yes	23	111,293	2.07	3.53	(2.28, 5.48) ***	1.75	(1.06, 2.88) *
hyperlipidemia							
No	183	2,875,027	0.64	1.00	(reference)		
Yes	6	49,573	1.21	1.90	(0.84, 4.30)		
CAD							
No	171	2,857,167	0.60	1.00	(reference)	1.00	(reference)
Yes	18	67,432	2.67	4.50	(2.76, 7.32) ***	1.59	(0.89, 2.84)
CVD							
No	176	2,874,382	0.61	1.00	(reference)	1.00	(reference)
Yes	13	50,218	2.59	4.25	(2.42, 7.48) ***	1.53	(0.80, 2.90)
CKD							
No	186	2,914,193	0.64	1.00	(reference)	1.00	(reference)
Yes	3	10,406	2.88	4.50	(1.44, 14.1) **	1.52	(0.46, 4.89)
COPD							
No	178	2,899,817	0.61	1.00	(reference)	1.00	(reference)
Yes	11	24,782	4.44	7.23	(3.93, 13.31) ***	2.65	(1.36, 5.16) **
HIV							
No	188	2,923,704	0.64	1.00	(reference)	1.00	(reference)
Yes	1	895	11.2	17.24	(2.41, 123.1) **	22.8	(3.18, 163.81) **
Liver cirrhosis							
No	185	2,916,568	0.63	1.00	(reference)	1.00	(reference)
Yes	4	8031	4.98	7.88	(2.92, 21.23) ***	4.48	(1.64, 12.25) **
SLE							
No	183	2,919,707	0.63	1.00	(reference)	1.00	(reference)
Yes	6	4893	12.3	19.3	(8.58, 43.63) ***	23.3	(10.20, 53.06) ***

Abbreviations: CAD: coronary artery disease; CVD: cerebrovascular disease; CKD: chronic kidney disease; COPD: chronic obstructive pulmonary disease; HIV: human immunodeficiency virus; SLE: systemic lupus erythematosus; *: *p*-value < 0.05; **: *p*-value < 0.01; ***: *p*-value < 0.001; CI, confidence interval; HR, hazard ratio; PY: person-years; IR, incidence rate per 10,000 person-years. ^†^: adjusted for age and all comorbidities except hyperlipidemia.

**Table 3 jcm-10-01466-t003:** Sensitivity Analyses.

	Compared to Patients without Appendectomy
	aHR (95% CI)
Model 1 (Main model)	1.61 (1.20, 2.17) **
Model 2	1.58 (1.17, 2.13) **
Model 3	1.61 (1.20, 2.16) ***
Model 4	1.71 (1.26, 2.33) ***
Model 5	1.24 (1.02, 1.52) *

Model 1: excluded patients with NTS occurred < 1 month after index date or PUD before index date or adjusted demographics and all comorbidities except hyperlipidemia in Table 1. Model 2: excluded patients with NTS occurred < 3 months after index date or PUD before index date. Model 3: excluded patients with recent bacterial infection within 6 months before the index date. Model 4: excluded patients with HIV, liver cirrhosis and SLE before the index date. Model 5: included patients with PUD into the main model analysis. aHR: adjusted hazard ratio; CI: confidence interval; *: *p*-value < 0.05; **: *p*-value < 0.01; ***: *p*-value < 0.001.

**Table 4 jcm-10-01466-t004:** Subgroup analysis.

	Appendectomy						Crude Analysis		Adjusted Analysis ^†^		
	No			Yes							*p* for Interaction
Variables	Events	PY	IR	Events	PY	IR	HR	95% CI	HR	95% CI	
Overall	77	1,402,999	0.55	112	1,521,600	0.74	1.35	(1.01, 1.8) *	1.61	(1.20, 2.17) **	
Gender											0.82
Female	36	683,410	0.53	60	743,265	0.81	1.56	(1.03, 2.36) *	1.92	(1.26, 2.93) **	
Male	41	719,589	0.57	52	778,335	0.67	1.16	(0.77, 1.75)	1.33	(0.88, 2.0)	
Age group											0.10
18–30	13	492,561	0.26	36	564,306	0.64	2.42	(1.28, 4.57) **	2.67	(1.41, 5.07) **	
31–40	13	328,638	0.4	23	377,947	0.61	1.57	(0.79, 3.10)	1.71	(0.86, 3.40)	
41–50	10	260,773	0.38	21	289,389	0.73	1.91	(0.90, 4.06)	2.05	(0.96, 4.38)	
51–60	4	147,585	0.27	11	153,000	0.72	2.62	(0.83, 8.24)	2.68	(0.85, 8.47)	
61–70	15	88,397	1.70	12	80,294	1.49	0.86	(0.40, 1.84)	1.12	(0.51, 2.45)	
71–80	12	64,375	1.86	7	44,753	1.56	0.82	(0.32, 2.09)	0.90	(0.34, 2.35)	
81–100	10	20,672	4.84	2	11,912	1.68	0.35	(0.08, 1.60)	0.34	(0.07, 1.63)	
Occupation											0.63
white-collar worker	38	739,180	0.51	64	829,610	0.77	1.52	(1.02, 2.27) *	1.76	(1.17, 2.65) **	
blue-collar worker	18	315,798	0.57	29	353,016	0.82	1.42	(0.79, 2.57)	1.69	(0.93, 3.09)	
farmer	2	29,308	0.68	1	30,566	0.33	0.48	(0.04, 5.30)	0.43	(0.04, 4.78)	
fisher	14	169,961	0.82	11	161,960	0.68	0.84	(0.38, 1.84)	1.03	(0.46, 2.32)	
others	5	148,751	0.34	7	146,449	0.48	1.41	(0.45, 4.46)	2.13	(0.64, 7.05)	
Comorbidities											
hypertension											0.15
No	53	1,269,090	0.42	99	1,445,802	0.68	1.65	(1.18, 2.30) **	1.77	(1.26, 2.47) ***	
Yes	24	133,909	1.79	13	75,798	1.72	0.92	(0.47, 1.82)	1.13	(0.57, 2.26)	
diabetes											0.18
No	62	1,335,540	0.46	104	1,477,765	0.7	1.52	(1.11, 2.08) **	1.72	(1.25, 2.37) ***	
Yes	15	67,459	2.22	8	43,835	1.83	0.83	(0.35, 1.97)	0.95	(0.4, 2.28)	
hyperlipidemia											0.20
No	72	1,371,739	0.52	111	1,503,288	0.74	1.41	(1.05, 1.90) *	1.65	(1.22, 2.23) **	
Yes	5	31,260	1.6	1	18,312	0.55	0.36	(0.04, 3.10)	0.49	(0.06, 4.31)	
CAD											0.48
No	65	1,357,680	0.48	106	1,499,487	0.71	1.48	(1.09, 2.02) *	1.68	(1.23, 2.3) **	
Yes	12	45,319	2.65	6	22,113	2.71	0.98	(0.37, 2.63)	1.09	(0.4, 2.97)	
CVD											0.76
No	69	1,370,138	0.5	107	1,504,243	0.71	1.42	(1.05, 1.92) *	1.63	(1.2, 2.21) **	
Yes	8	32,861	2.43	5	17,357	2.88	1.16	(0.38, 3.55)	1.42	(0.45, 4.54)	
CKD											0.44
No	76	1,396,400	0.54	110	1,517,792	0.72	1.34	(1.00, 1.79)	1.59	(1.18, 2.15) **	
Yes	1	6599	1.52	2	3808	5.25	3.66	(0.33, 40.56)	3.68	(0.32, 42.6)	
COPD											0.13
No	68	1,386,525	0.49	110	1,513,293	0.73	1.49	(1.1, 2.01) *	1.71	(1.26, 2.33) ***	
Yes	9	16,475	5.46	2	8307	2.41	0.46	(0.1, 2.11)	0.46	(0.1, 2.21)	
HIV											0.97
No	76	1,402,535	0.54	112	1,521,169	0.74	1.37	(1.02, 1.83) *	1.63	(1.21, 2.19) **	
Yes	1	465	21.52	0	431	0	0	(0, Inf)	0	(0, 0.)	
Liver cirrhosis											0.81
No	75	1,397,922	0.54	110	1,518,646	0.72	1.36	(1.01, 1.82) *	1.62	(1.2, 2.18) **	
Yes	2	5077	3.94	2	2954	6.77	1.84	(0.26, 13.09)	1.51	(0.2, 11.33)	
SLE											0.36
No	72	1,399,487	0.51	111	1,520,219	0.73	1.43	(1.06, 1.92) *	1.67	(1.24, 2.26) ***	
Yes	5	3512	14.24	1	1381	7.24	0.5	(0.06, 4.32)	0.49	(0.06, 4.22)	

Abbreviations: CAD: coronary artery disease; CVD: cerebrovascular disease; CKD: chronic kidney disease; COPD: chronic obstructive pulmonary disease; HIV: human immunodeficiency virus; SLE: systemic lupus erythematosus; *: *p*-value < 0.05; **: *p*-value < 0.01; ***: *p*-value < 0.001; Abbreviations: CI, confidence interval; HR, hazard ratio; PY: person-years; IR, incidence rate per 10,000 person-years. ^†^: adjusted for demographics and all comorbidities except hyperlipidemia.

**Table 5 jcm-10-01466-t005:** Incidence and hazard ratios for NTS infection according to follow-up year.

	Appendectomy						Crude Analysis		Adjusted Analysis ^†^	
	No			Yes						
Follow up Time	Events	PY	IR	Events	PY	IR	HR	95% CI	HR	95% CI
<6 months	8	103,881	0.77	12	103,674	1.16	1.50	(0.61, 3.68)	1.83	(0.74, 4.53)
6 months–1 year	11	103,081	1.07	6	102,896	0.58	0.55	(0.20, 1.48)	0.68	(0.25, 1.89)
>1 year	58	1,196,037	0.48	94	1,315,030	0.71	1.47	(1.06, 2.05) *	1.74	(1.25, 2.43) **

*: *p*-value < 0.05; **: *p*-value < 0.01; Abbreviations: CI, confidence interval; HR, hazard ratio; PY: person-years; IR, incidence rate per 10,000 person-years. ^†^: adjusted for demographics and all comorbidities except hyperlipidemia.

**Table 6 jcm-10-01466-t006:** Negative Control Exposure—Incidence Rate and Hazard Ratio.

	NTS						
Variables	Events	PY	IR	cHR	95% CI	aHR	95% CI ^†^
Diverticulitis							
No	4	14,099	2.84	1.00	-	1.00	-
Yes	3	13,858	2.16	0.78	(0.17, 3.47)	0.85	(0.18, 3.95)

^†^: adjusted for demographics and all comorbidities except hyperlipidemia.

## Data Availability

Data are available from the National Health Insurance Research Database (NHIRD) published by Taiwan National Health Insurance (NHI) Bureau. Due to legal restrictions imposed by the government of Taiwan in relation to the “Personal Information Protection Act”, data cannot be made publicly available.

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
