# Peer review of "Appendectomy and Non-Typhoidal Salmonella Infection: A Population-Based Matched Cohort Study"

_jcm, 2021, doi:10.3390/jcm10071466_

Round 1
Reviewer 1 Report
In the article entitled “Appendectomy and Non-typhoidal Salmonella Infection: A Population-based Matched Cohort Study” authors Wu et al present results of an original investigation examining the association between appendectomy and non-typhoidal salmonella infection. They identified over 208 thousand patients who underwent appendectomy between 2000 and 2012 using the Taiwan National Health Insurance Research Database, and compared these with a cohort of propensity matched patients who did not undergo appendectomy during a similar time period. Their outcome of interest was hospitalization for NTS infection. They found that patients who had undergone appendectomy had significantly higher risk of subsequent NTS hospitalization. The authors suggest that the development of NTS infection requiring hospitalization may be related to the disruption of microbiome after appendectomy. The authors should be commended on presenting a thorough investigation. However, there area a few questions that must be addressed:
- The authors excluded patients who had PUD as proxy for PPI because PPIs have been shown to be associated with higher risk of NTS infection – however, for other comorbidities related to immunocompromise that also confer increased risk of NTS, these were adjusted for in PSM and in the multivariable regression model. Why was PUD treated differently (ie pts excluded rather than adjusting for PUD as a comorbidity)
- Were patients who underwent multiple concurrent procedures at the index date excluded? The CPT code for appendectomy can be listed along with other procedures – patients who underwent more than one procedure at time of appendectomy should also be excluded from the patient cohort.
- Was the length of antibiotic treatment captured by the database? Although the authors did not include episodes of NTS within 30 days of index date to account for potential microbiome disruption related to antibiotic treatment, literature has shown that the effects of antibiotic treatment is often much longer lasting than 30 days
- Did the authors collect data on any antibiotic treatment for the patient cohorts between index date and event/ end date? Antibiotic use is likely a significant confounder if the main hypothesis is that disruptions in microbiome drives increased susceptibility to NTS.
Overall, while the authors have made a clear presentation and robust statistical analysis, their ultimate conclusion seems questionable – without sufficient strong prior evidence that removal of the appendix either significantly alters the microbiome or affects a patient’s immune function, the leap between appendectomy to development of NTS seems a little too far of a reach and the language should be amended. A major weakness of the study is the absence of data on use of antibiotics in both cohorts, which is likely a significant confounder. Additionally, the clinical relevance of their findings is unclear, as overall the incidence of NTS is quite low, and the authors do not comment on the practical clinical implications of their findings.
Author Response
In the article entitled “Appendectomy and Non-typhoidal Salmonella Infection: A Population-based Matched Cohort Study” authors Wu et al present results of an original investigation examining the association between appendectomy and non-typhoidal salmonella infection. They identified over 208 thousand patients who underwent appendectomy between 2000 and 2012 using the Taiwan National Health Insurance Research Database, and compared these with a cohort of propensity matched patients who did not undergo appendectomy during a similar time period. Their outcome of interest was hospitalization for NTS infection. They found that patients who had undergone appendectomy had significantly higher risk of subsequent NTS hospitalization. The authors suggest that the development of NTS infection requiring hospitalization may be related to the disruption of microbiome after appendectomy. The authors should be commended on presenting a thorough investigation. However, there area a few questions that must be addressed:
- The authors excluded patients who had PUD as proxy for PPI because PPIs have been shown to be associated with higher risk of NTS infection – however, for other comorbidities related to immunocompromise that also confer increased risk of NTS, these were adjusted for in PSM and in the multivariable regression model. Why was PUD treated differently (ie pts excluded rather than adjusting for PUD as a comorbidity)
Response: We thank you very much for giving us the opportunity to revise it.
[In Tacking Version, Line 172-176]
We have rewritten as follow:
In model 4, PUD and other comorbidities related to immunocompromise that confer increased risk of NTS infection were excluded before the index date.
In model 5, PUD and other comorbidities related to immunocompromise that confer increased risk of NTS infection were not excluded and adjusted as covariates in regression analysis.
[In Tacking Version, Line 218-227]
We have rewritten as follow:
Table 3 provides five models to examine the stability of HR of hospitalized NTS infection in different definition of appendectomy exposure and events of main outcome. In the subset of NHIRD (hospitalization dataset analyzed in this study), there was no prescription information to identify the length of antibiotic treatment. We developed 1, 2, and 3 to minimize the effect of antibiotics utility. The wash-out period in model 2 was up to 90 days. The aHRs were 1.61 (95% CI 1.20 to 2.17), 1.58 (95% CI 1.17 to 2.13) and 1.61 (95% CI 1.20 to 2.16) in models 1, 2 and 3. Furthermore, in model 4, we excluded immunocompromised cases potentially prone to have NTS infection and the aHR was 1.71 (95% CI 1.26 to 2.33) of NTS infection for appendectomy exposure. In model 5, PUD was included into regression analysis and the aHR was 1.24 (95% CI 1.02 to 1.52).
- Were patients who underwent multiple concurrent procedures at the index date excluded? The CPT code for appendectomy can be listed along with other procedures – patients who underwent more than one procedure at time of appendectomy should also be excluded from the patient cohort.
Response: All authors thank the reviewer for indicating the unclear point and giving us opportunity to clarify it.
[In Tacking Version, Line 99-100]
The manuscript has been rewritten as follows:
To minimize confounding from other alimentary surgical procedures, individuals who received a gastrectomy (ICD-9-CM procedure code 43.5-43.9), cholecystectomy (ICD-9-CM procedure code 51.2), or intestinal or large bowel resection (ICD-9-CM procedure code 45.6-45.9) before the index date or underwent multiple concurrent procedures at time of appendectomy were excluded.
- Was the length of antibiotic treatment captured by the database? Although the authors did not include episodes of NTS within 30 days of index date to account for potential microbiome disruption related to antibiotic treatment, literature has shown that the effects of antibiotic treatment is often much longer lasting than 30 days
Response:
We thank you very much for reviewing articles carefully and giving us the opportunity to revise it.
[In Tacking Version, Line 164-169]
Because the effects of antibiotics treatment might be longer lasting than 30 days; in model 2, we excluded patients having NTS infection occurred 0 to 90 days after the index date. In model 3, because antibiotics utilization is likely a significant confounder, and no prescription information in the hospitalization dataset, we excluded patients having bacterial infection within 6 months before the index date (a proxy to minimize the effect of antibiotics utilization on the participants).
[In Tacking Version, Line 218-223]
Table 3 provides five models to examine the stability of HR of hospitalized NTS infection in different definition of appendectomy exposure and events of main outcome. In the subset of NHIRD (hospitalization dataset analyzed in this study), there was no prescription information to identify the length of antibiotic treatment. We developed 1, 2, and 3 to minimize the effect of antibiotics utility. The wash-out period in model 2 was up to 90 days. The aHRs were 1.61 (95% CI 1.20 to 2.17), 1.58 (95% CI 1.17 to 2.13) and 1.61 (95% CI 1.20 to 2.16) in models 1, 2 and 3.
- Did the authors collect data on any antibiotic treatment for the patient cohorts between index date and event/ end date? Antibiotic use is likely a significant confounder if the main hypothesis is that disruptions in microbiome drives increased susceptibility to NTS.
Response: All authors thank you for your constructive comments.
We have mentioned the limitation in our revised version.
[In Tacking Version, Line 166-169]
…because antibiotics utilization is likely a significant confounder, and no prescription information in the hospitalization dataset, we excluded patients having bacterial infection within 6 months before the index date (a proxy to minimize the effect of antibiotics utilization on the participants).
[In Tacking Version, Line 219-222]
…In the subset of NHIRD (hospitalization dataset analyzed in this study), there was no prescription information to identify the length of antibiotic treatment. We developed 1, 2, and 3 to minimize the effect of antibiotics utility.
[In Tacking Version, Line 358]
…Finally, microbial dysbiosis may be a key intermediate between appendectomy and subsequent NTS infection, but that has not been established in current study for lacking detailed information regarding the interactions between appendectomy, change of microbiome metabolites (short-chain fatty acids, such as acetate, propionate, and butyrate), and the information of antibiotic use.
Overall, while the authors have made a clear presentation and robust statistical analysis, their ultimate conclusion seems questionable – without sufficient strong prior evidence that removal of the appendix either significantly alters the microbiome or affects a patient’s immune function, the leap between appendectomy to development of NTS seems a little too far of a reach and the language should be amended. A major weakness of the study is the absence of data on use of antibiotics in both cohorts, which is likely a significant confounder. Additionally, the clinical relevance of their findings is unclear, as overall the incidence of NTS is quite low, and the authors do not comment on the practical clinical implications of their findings.
Response: All authors thank you for your comments. We have toned down and rewritten the sentences as follow:
[In Tacking Version, Line 362-364]
A small number of NTS infection diagnoses occurred in the study, thus limiting the conclusions somewhat. Clinicians are advised to implement prudently on the post-operation education for patients to get rid of possible NTS contaminated food in endemic area.
[In Tacking Version, Line 60-62]
We have cited a recent study that enrolled patients underwent incidental prophylactic appendectomy during 2004-2008 showed profound and long-term dysbiosis in these patients up to years [6]
All authors thank you very much.
We have learned a great deal from the comments You provided.
The English Editing has been made. Thank you.

Reviewer 2 Report
The study is well conducted, with solid statistical analysis.
I would appreciate it if the authors explained which would be the potential and practical improvements deriving from their findings.
Thank you.
Author Response
Reviewer 2
The study is well conducted, with solid statistical analysis.
I would appreciate it if the authors explained which would be the potential and practical improvements deriving from their findings. Thank you.
Response: All authors thank you for your comments. We have added some sentences for potential and practical improvements deriving from our findings as follow:
[In Tacking Version, Line 362-364]
A small number of NTS infection diagnoses occurred in the study, thus limiting the conclusions somewhat. Clinicians are advised to implement prudently on the post-operation education for patients to get rid of possible NTS contaminated food in endemic area.
All authors thank you very much.

Reviewer 3 Report
"Appendectomy and Non-typhoidal Salmonella Infection: A Population-based Matched Cohort Study" is an interesting paper. Sample size are wide and it allowed authors to make several adjustments and sub-analysis. I have found curious that the risk of hospitalized NTS increased with tha passage of time from appendectomy. Overall a good work.
Author Response
Reviewer 3
"Appendectomy and Non-typhoidal Salmonella Infection: A Population-based Matched Cohort Study" is an interesting paper. Sample size are wide and it allowed authors to make several adjustments and sub-analysis. I have found curious that the risk of hospitalized NTS increased with tha passage of time from appendectomy. Overall a good work.
Response: All authors thank you for your comments.
The cause of increased risk of hospitalized NTS infection along with the passage of time from appendectomy cannot be made from our observation study. We have toned down our language and re-written as follow:
[In Tacking Version, Line 362-363]
A small number of NTS infection diagnoses occurred in the study, thus limiting the conclusions somewhat.
[In Tacking Version, Line 364-366]
It is of noted that since this observation study was performed from one, relatively small country, if similar studies will be done in the future in other countries of non-Asian origin, the results may be exactly the opposite.
All authors thanks for your comments.

Round 2
Reviewer 1 Report
The authors have satisfactorily responded to all comments.
Reviewer 2 Report
Thank you for the short, but significant, clinical advice.